# Association Between *BRAF V600E* Allele Frequency and Aggressive Behavior in Papillary Thyroid Microcarcinoma

**DOI:** 10.3390/cancers17152553

**Published:** 2025-08-01

**Authors:** Luiza Tatar, Saruchi Bandargal, Marc P. Pusztaszeri, Véronique-Isabelle Forest, Michael P. Hier, Jasmine Kouz, Raisa Chowdhury, Richard J. Payne

**Affiliations:** 1Faculty of Medicine and Health Sciences, McGill University, Montreal, QC H3A 0G4, Canada; 2Department of Pathology, Jewish General Hospital, McGill University, Montreal, QC H3T 1E2, Canada; 3Department of Otolaryngology—Head and Neck Surgery, Jewish General Hospital, Montreal, QC H3T 1E2, Canada; 4Department of Endocrinology, Montreal Sacred Heart Hospital, Montreal, QC H4J 1C5, Canada; 5Department of Otolaryngology—Head and Neck Surgery, McGill University Health Centre, Montreal, QC H4A 3J1, Canada

**Keywords:** thyroid cancer, papillary thyroid microcarcinoma, molecular testing, *BRAF V600E*, allele frequency

## Abstract

Papillary thyroid microcarcinoma (MPTC) is increasingly detected due to advanced imaging techniques. While most cases are slow-growing and treated conservatively, some show aggressive behavior, making treatment decisions challenging. This study focuses on the role of a genetic mutation, *BRAF V600E*, and its allele frequency (the proportion of mutated DNA) in predicting the aggressiveness of these cancers. By analyzing data from patients with MPTC, the researchers aim to determine if a higher allele frequency is linked to more aggressive disease features. If confirmed, these findings could help inform clinical decisions regarding active surveillance or surgery for patients with MPTC, improving personalized care and treatment outcomes.

## 1. Introduction

The global incidence of thyroid carcinoma has risen significantly, increasing from 4.9 per 10,000 individuals in 1975 to 10.1 per 100,000 women and 3.1 per 100,000 men by 2020 [1,2,3]. Despite this growth, thyroid cancer mortality rates have remained relatively stable, a trend attributed to the enhanced detection of papillary thyroid microcarcinomas (MPTCs) [3,4]. Often, MPTCs are found using ultrasonography, or are incidental discoveries on unrelated imaging investigations [3,4]. MPTC, a subset of papillary thyroid carcinoma (PTC), constitutes the most common form of thyroid malignancy, with PTC accounting for approximately 85% of all thyroid cancers [5,6]. According to the 2015 American Thyroid Association Guidelines, MPTC is defined as thyroid nodules measuring ≤ 1 cm in diameter [7].

MPTC nodules are generally associated with a low morbidity and excellent survival due to their slow progression [4]. These nodules have been detected in up to 35.6% of autopsy studies on individuals with a healthy thyroid, and are thought to develop during childhood and adolescence, becoming quiescent in early adulthood [8]. However, 19% of patients with MPTC present with aggressive features such as lymph node metastasis, extrathyroidal extension or lymphovascular invasion [9]. Thus, a non-negligible subset of MPTC will exhibit aggressive patterns on post-operative and histological specimens [10]. In PTC, both tall cell and hobnail subtypes, associated with worse patient prognosis, have been linked with *BRAF V600E*-positive thyroid nodules [11]. The current management of MPTC nodules is highly individualized. Detection of prognostic factors such as the *BRAF V600E* mutation through molecular testing guides clinicians to reliably target patients in need of more extensive surgical and medical therapy [12].

*BRAF* is a proto-oncogene situated on the long arm of chromosome 7 (7q34) [13]. The point mutation *V600E* occurs when glutamic acid (E) substitutes valine (V) on codon 600. The mutation then constitutively activates the BRAF kinase which upregulates the mitogen-activated protein kinase (MAPK) signaling pathway [13,14]. This causes uncontrolled cell proliferation and subsequent carcinogenesis [14]. Physicians recommend molecular testing, relying on next-generation sequencing to identify an ever-increasing number of mutations per DNA sample, including *BRAF V600E*. This mutation is the most frequently identified genetic alteration in PTC and has been implicated in increased tumor aggressiveness. *BRAF V600E* is associated with adverse features, including tall cell histological subtypes, macroscopic extrathyroidal extension (ETE), lymph node metastasis, advanced tumor stages, and higher recurrence rates [9,10,14,15,16,17,18]. A hyperactivated MAPK pathway is linked to poor plasma membrane uptake in iodine, explaining the reduced therapeutic effectiveness of adjuvant radioactive iodine (I-131) administration [19]. Contradictory literature exists surrounding *BRAF V600E* as an accurate reflection of the malignancy and prognosis of thyroid tumors. For example, despite highlighting increased rates of *BRAF V600E* mutation in PTC with hobnail features, Spyroglu et al. showed that *BRAF V600E* negativity does not improve patient survival [11].

A critical genetic testing parameter that may provide further insight in the pathophysiological development of MPTC is allele frequency (AF). AF is a genetic measure quantifying the proportion of mutated DNA relative to wild-type DNA in a given sample [11,16,20].

In PTC, elevated *BRAF V600E* AF has been linked to aggressive features such as sentinel lymph node positivity [16,17,21], extensive ETE, and increased rates of metastasis, contributing to advanced disease staging and poorer clinical outcomes [20,22]. Although the relationship between elevated *BRAF V600E* AF and aggressiveness in PTC is well-documented, its role in MPTC remains uncertain. With the rising incidence of incidentally discovered thyroid micronodules, there is a growing need for reliable histopathological markers to assess tumor aggressiveness. Investigating *BRAF V600E* AF in MPTC may provide critical insights, potentially informing clinical decisions regarding active surveillance versus surgical intervention and determining the extent of surgical management required.

The main objective was to explore the association between *BRAF V600E* AF and the presence of any aggressive histopathological features in MPTC tumors. The secondary objective was to explore the associations between *BRAF V600E* AF and individual aggressive histopathological features in MPTC tumors. We hypothesized that higher *BRAF V600E* AF is positively correlated with increased tumor aggressiveness in MPTC, as a greater proportion of mutated DNA may drive oncogenic processes, leading to more aggressive clinical and pathological features.

## 2. Materials and Methods

Patients aged 18 years or older who tested positive for the *BRAF V600E* mutation on pre-operative molecular testing and had a thyroid nodule measuring ≤ 1.0 cm in diameter, as confirmed by the final pathological sample, were included in the study. Exclusion criteria included patients without AF testing or cases where the thyroid nodule biopsied from molecular testing was not highlighted as main finding of the pathological report. Patient data from 1 January 2016 to 23 December 2024, were retrieved from two McGill University tertiary care teaching hospitals: Jewish General Hospital and McGill University Health Center databases.

The following data were collected: patient age, sex, nodule location, Bethesda score, molecular testing results using ThyroSeqV3 (CBLPath, Inc., Rye Brook, NY, USA), tumor dimensions (longest axis on ultrasound and in the final pathological sample), presence of lymph node metastasis, extrathyroidal extension (ETE), extensive lymphovascular invasion (LVI), and histological subtype (e.g., classical, oncocytic, follicular, columnar cell, diffuse sclerosing, hobnail, solid, or tall cell). Diagnoses from fine-needle aspiration (FNA) biopsy samples were classified using the Bethesda System for Reporting Thyroid Cytology [23,24].

Pathological diagnoses were based on the 2017 World Health Organization (WHO) classification of endocrine tumors [24]. Cytology and final pathology samples were independently reviewed by board-certified, head and neck fellowship-trained pathologists.

To ensure confidentiality, all participants’ personal information was coded and securely protected during data collection and analysis. Ethical approval was granted by both the McGill University Health Center and CIUSSS West-Central Research Ethics Board in Montreal, QC, Canada (MP-05-2024-4145).

### 2.1. Statistical Analyses

#### Variables

Tumor aggressiveness was categorized as aggressive or non-aggressive. Nodules were classified as aggressive if they exhibited one or more of the following features: lymph node metastasis, ETE, extensive LVI, or high-risk histological subtypes (solid/trabecular, columnar cell, diffuse sclerosing, hobnail, or tall cell) [23]. Classical subtypes of PTC with >15% tall cell features were included in the tall cell group due to their documented increased aggressiveness [25]. Non-aggressive nodules included those with classical, oncocytic, or follicular subtypes of PTC [16].

Allele frequency was treated as a continuous variable. Individual histo-pathological determinants of MPTC aggressiveness (i.e., lymph node metastasis, macroscopic extrathyroidal extension, extensive lymphovascular invasion, and histological subtype aggressiveness) were coded as binary variables (yes or no) based on the presence or absence of metastasis, extrathyroidal or lymphovascular extension, or an aggressive/non-aggressive histological subtype.

### 2.2. Analysis Methods

In line with the main objective, we used an independent two-sample *t*-test to assess differences in *BRAF V600E* AF by tumor aggressiveness. For the secondary objective, we applied the same statistical approach to compare AF between groups defined by lymph node metastasis and histological subtype aggressiveness. To ensure the validity of our results, we checked the following assumptions: (a) Normal distribution of AF using the Anderson-Darling test. We assumed AF was normally distributed if *p* > 0.05, and (b) Homogeneity of variance using Levene’s test. Group variances in AF were considered equal if *p* > 0.05. Descriptive statistics included mean (M) and standard deviation (SD) for AF and proportions (percentages) for binary variables. The results were visually presented using boxplots, which illustrate the distribution of AF, including the median and interquartile range. Specifically, the boxplots display the AF by tumor aggressiveness (non-aggressive vs. aggressive; Figure 1), lymph node metastasis status (negative vs. positive; Figure 2) and histologic subtype (non-aggressive vs. aggressive; Figure 3). In each of the three figures, horizontal lines indicating the mean AF of each subgroup are superimposed on the boxplots. All analyses were conducted using the R statistical software, version 4.3.3.

## 3. Results

### 3.1. General Findings

A total of 148 cases met the inclusion criteria among 1564 patient charts reviewed. 85 were excluded because thyroid nodule AF was not tested, 29 were excluded because the nodule for which molecular testing was obtained did not correspond to the main nodule described by pathologists in the final report. 34 observations met the inclusion criteria and were retained in the final analysis. The mean AF was 19.24 (SD 13.13) (Table 1). Lymph node metastases were found in 10 individuals (29.4%), and 7 individuals (20.6%) had aggressive histological subtypes (Table 1). The Alderson-Darling test indicated a normal distribution of AF in the analyzed sample (N = 34, *p* = 0.06). Levene’s test confirmed equal variance of AF between groups across all conducted subgroup analyses (*p* > 0.05).

### 3.2. Association Between BRAF V600E AF and Any Histo-Pathological Determinant of MPTC Tumor Aggressiveness

Corresponding to the main objective, we found a significantly higher mean (M) AF in individuals with aggressive tumors (M = 23.58) compared to those with non-aggressive tumors (M = 13.73) (95% CI: −18.53; −1.16; t = −2.31; df = 32; *p* = 0.03) (Figure 1).

### 3.3. Associations Between BRAF V600E AF and Individual Histo-Pathological Determinants of MPTC Tumor Aggressiveness

In line with the secondary objectives, the differences in mean AF by lymph node metastasis status (M no metastasis = 16.67; M metastasis = 25.4) (Figure 2) and histological subtype (M aggressive = 19.57; M not aggressive = 19.15) (Figure 3) were not statistically significant (t = −1.83; df = 32; *p* = 0.08 and t = −0.07; df = 32; *p* = 0.94, respectively). Although not statistically significant at the *p* < 0.05 threshold, a trend was observed suggesting a higher AF in cases with lymph node metastasis (Figure 2). Due to the limited number of observations for macroscopic extrathyroidal extension (n = 3) and extensive lymphovascular invasion (n = 1), subgroup analyses were not feasible.

### 3.4. Supplementary Analyses

Post hoc analyses considering a calculated medium effect size (Cohen’s d = 0.79) for the difference in AF by aggressiveness, unequal group sizes (n1 = 15; n2 = 19) and a significance level of α = 0.05 revealed a power of 0.61. For a future study, a minimum of 26 participants per group (N = 52 in total) would be required to detect a medium effect size (d = 0.79) using a two-sided *t*-test with 80% power and α = 0.05. For AF by lymph node metastasis, the power was 0.43 based on unequal group sizes (n1 = 10, n2 = 24), d = 0.69 and α = 0.05. To achieve 80% power, we estimated a minimum required sample size of 34 participants per group (N = 68 in total).

**Figure 1 cancers-17-02553-f001:**
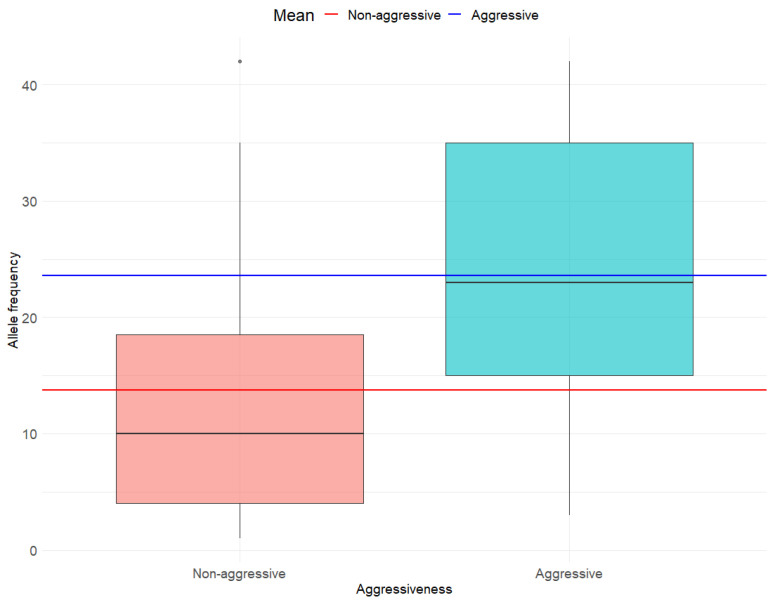
Allele frequency by tumor aggressiveness.

**Figure 2 cancers-17-02553-f002:**
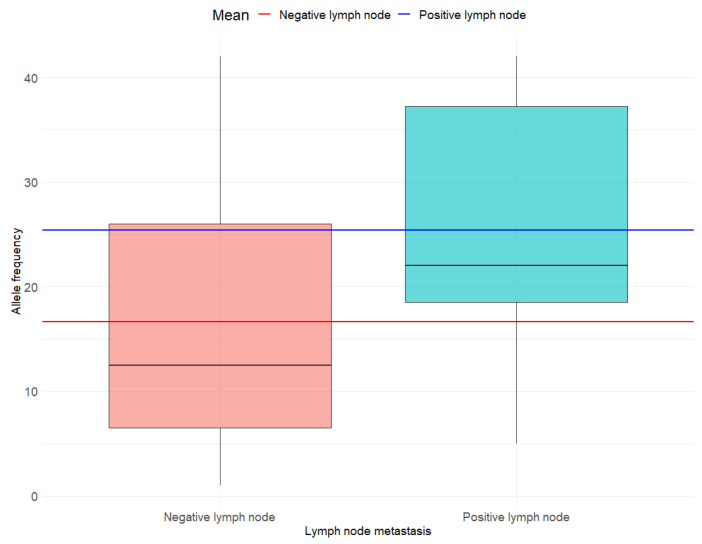
Allele frequency by lymph node metastasis.

**Figure 3 cancers-17-02553-f003:**
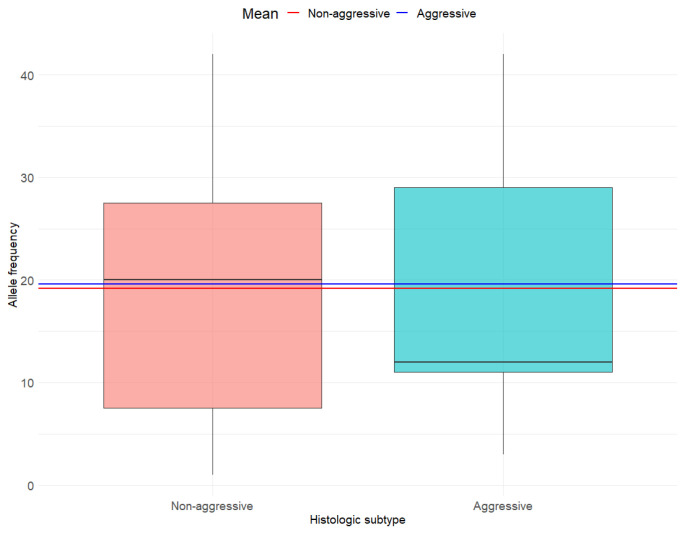
Allele frequency by histologic subtype.

## 4. Discussion

The relationship between the *BRAF V600E* mutation and aggressive clinicopathological features in thyroid cancer, particularly in MPTC, remains a subject of debate, with studies reporting mixed findings. For instance, Xie et al. identified factors such as age and capsular invasion—rather than the *BRAF V600E* mutation—as stronger predictors of lymph node metastases in MPTC [26]. Similarly, Chen et al. found no significant correlation between the mutation and variables such as the number of lymph node metastases, extranodal extension, or overall disease stage [27]. In contrast, Lai et al. found that *BRAF V600E* is associated with lymph node metastasis and bilateral thyroid carcinoma in MPTC nodules [28]. Attia et al. demonstrated a positive association between the mutation and aggressive features like tumor multifocality, ETE, and lymph node metastases [9]. These conflicting results underscore the need to explore additional factors, such as *BRAF V600E* AF, to clarify its potential role in predicting tumor aggressiveness.

### 4.1. Aggressivity

In this study, we investigated whether elevated *BRAF V600E* AF is associated with aggressive histopathological features in MPTC. Our findings reveal that individuals with aggressive MPTC exhibited significantly higher mean *BRAF V600E* AF compared to those with non-aggressive MPTC. *BRAF V600E* is a clonal mutation, arising during initial phases of the DNA modifications [29]. It characteristically drives rapid cellular division and confers adaptive advantages to oncotic cells [29]. Because AF represents allelic incidence across multiple loci, we can infer that increased mean AF contributes to mutation load, promoting the exponential translation of the mutant *BRAF* kinase and subsequent aberrant activation of the MAPK pathway [16]. This wayward activation may promote the rapid emergence of aggressive histopathological features because of uncontrolled proliferation [14]. In effect, the literature has demonstrated that PTC aggressiveness is linked to *BRAF V600E* on molecular testing [14,16,17]. Silver et al. found that *BRAF V600E*-positive MPTC nodules had one or more aggressive histopathological feature [10]. This evidence, coupled with our findings, lead us to believe that MPTC nodules with a high AF have increased aggressive behavior.

### 4.2. Lymph Node Metastasis

While our results indicate a strong association between elevated *BRAF V600E* AF and overall tumor aggressiveness, no statistically significant correlations were observed between AF and lymph node metastasis. We only identified a trend between these two variables. In contrast, the literature shows a well-established association between increased mean *BRAF V600E* AF and lymph node metastasis in PTC [9,10,14,16,17,18,21]. While limited literature linked *BRAF V600E* AF to lymph node metastasis, a meta-analysis conducted by Wang et al. showed that primary MPTCs with a maximal diameter 0.5 cm–1 cm are associated with lymph node metastasis, suggesting that size plays an important role in invasion [30]. Tallini et al. reported similar findings, showing that *BRAF V600E*-positive MPTC are linked to a subcapsular site of origin, thereby increasing the likelihood of developing aggressive anatomopathological features including lymph node metastases [31]. Contrary to the existing literature, our data does not support AF as a predictor of LN in *BRAF V600E*-positive MPTC nodules. Our results suggest that beyond *BRAF V600E* AF, genetic variability, age and ethnicity may contribute to the heterogenous rate of carcinogenic cell proliferation, tumor growth, and subsequent lymph node metastasis [30,32]. We may have not considered enough MPTC nodules to replicate a significant association between *BRAF V600E* AF and ETE.

### 4.3. ETE

We did not observe a statistically significant association between ETE and *BRAF V600E* AF. The literature suggests that *BRAF V600E* is associated with ETE in both PTC and MPTC [9,10,14,15,16,17,18,30,32]. ETE manifests with large, subcapsular thyroid nodules characterized by a greater distance separating the center of the tumor and the thyroid capsule [31]. Tallini et al. found that 95% of aggressive MPTC thyroid nodules with ETE are located on average 3 mm from the thyroid capsule [31]. These tumors were associated with the *BRAF V600E* mutation, and ETE [31]. Their increased anatomical proximity to the nearby musculature, the trachea, and the laryngeal nerve is believed to favor ETE formation [30,31]. Mutations aside, the distance between the tumor center and the surface of the thyroid capsule follows a Bell-curve distribution, suggesting that MPTC nodule development is random [31]. However, according to Tallini et al. *BRAF V600E*-positive MPTCs are associated with greater tumor center-capsule distance and a subcapsular origin, thereby increasing ETE development likelihood [31].

AF’s role in ETE remains poorly understood. Adbulhaleem et al. did not find an association between mean *BRAF V600E* AF and ETE [16]. Perhaps *BRAF V600E* AF becomes a significant contributor to ETE only in the presence of synergistic factors to thyroid carcinoma progression, such as aggressive histological subtype and lymph node metastases [9]. The non-association we observed may be explained by the small sample size of MPTC nodules. Factors other than AF may contribute to heterogenous tumor growth rates, and interpersonal variability. Also, the patients included in our study were under active surveillance, meaning that resected nodules were likely still in the early stages of pathogenesis, before the manifestations of the previously discussed synergistic effect.

### 4.4. Histological Subtype and Extensive LVI

Although prior studies have linked higher *BRAF V600E* AF with the tall cell histological subtype and extensive LVI in PTC, we cannot report a similar association in MPTC nodules [9,10,14,16,17,18,21]. As *BRAF V600E* confers aggressive progression, it is believed to be associated with the tall cell and hobnail histological subtypes [9,10,11,14,16,17,18,21]. Extensive LVI is believed to be influenced by tumor size and location, through similar pathophysiologic principles as those discussed for lymph node metastasis and ETE [9,30,31].

### 4.5. Prognosis

Accurate risk stratification for MPTC nodules is critical because despite the indolent majority, a subset of nodules will develop aggressive malignant features, local and distant metastases [10]. The 2015 American Thyroid Association guidelines recommend FNA for nodules ≥ 1 cm if an intermediate or highly suspicious pattern is seen on ultrasound [7]. According to recommendation 12 of their rapport, surgery is not an acceptable therapy for MPTC patients with comorbidities, short remaining lifespans, other medical issues requiring medical attention before thyroid surgery and without high-risk features on ultrasound [7]. A lobectomy should be considered for MPTC patients with previous exposure to ionizing radiation or with a family history of thyroid carcinoma, with a positive FNA result or with nodules exhibiting high-risk features (i.e., ETE or lymph node metastasis) [7]. Patients who are not candidates for surgery undergo active surveillance.

Prospective studies conducted in Japanese centers have shown that the disease-specific mortality and loco-regional recurrence rates are the same for patients having undergone an immediate lobectomy and those being actively surveilled [33,34]. However, Wang et al. recently conducted a meta-analysis showing that PTC and MPTC are prone to an early onset of lymph node metastasis, independently of *BRAF* mutation status. Furthermore, it is well-established that no clinical or radiologic features will specifically identify aggressive features, among which lymph node metastases and ETE [14]. Preoperative imaging such as ultrasound and ETE has a detection rate between 28 and 39% for central lymph node metastasis, the principal site of metastases for PTC and MPTC [30]. As a result, 30–65% of lymph node metastases are found postoperatively through pathologic and histologic analyses [14]. Positive lymph nodes and ETE discovered at this time are a marker of poor prognosis and are associated with decreased survival and increased recurrence rate. The controversy surrounding MPTC nodules highlights the need for an accurate prognostic factor, such as *BRAF V600E* AF status, to further inform and guide management. Our findings suggest that *BRAF V600E* AF may be an indicator of MPTC overall aggressivity, but they do not support this measure as a reflection of the individual determinants of aggressivity or as a prognostic factor.

The *BRAF V600E* AF has recently been implicated in the metastatic potential of PTC, with studies suggesting that higher AF is associated with poorer clinical outcomes and advanced tumor staging due to increased invasiveness and lymph node metastases [12]. Schumm et al. reported a that disease persistence and recurrence was associated with a higher median AF, and poorer recurrence-free survival [35]. *BRAF V600E* positivity alone as a prognostic factor for PTC and MPTC is extensively documented in the literature. Li et al. reported that, in PTC, *BRAF V600E* is associated with increased cancer-related mortality and increased recurrence. In MPTC nodules, they found that 4 to 13% of patients with a *BRAF V600E*-positive thyroid nodule will recur [14]. A meta-analysis conducted by Attia et al. highlighted that *BRAF V600E*-positive MPTC nodules are more resistant to radioactive iodine ablation therapy and have an increased risk of disease recurrence within a 5 year period [9]. Nechifor-Boila et al. suggest that *BRAF V600E* in PTC acts synergistically with aggressive pathological features such as extrathyroidal extension pT3b tumor stage, lymph node metastasis, positive surgical resection margins, persistent disease status and distant metastases [36]. The authors suggested *BRAF V600E* status be completed with radiologic findings to determine patient’s prognosis. However, contradictory evidence exists, with some studies reporting no significant impact of *BRAF V600E* AF on the prognosis or invasiveness of PTC [37,38]. These discrepancies may arise from several factors, including population diversity, geographic variations, and differences in the risk stratification and clinical management of patients across studies [20].

The utility of *BRAF V600E* AF as a prognostic marker depends on patient risk profiles. In low-risk thyroid cancer patients, the prognostic significance of AF may be less apparent, potentially contributing to inconsistencies in the literature. In contrast, its association with tumor aggressiveness and clinical outcomes becomes more distinct in intermediate-to-high-risk patients, where higher AF is consistently linked to more aggressive disease characteristics. This variability underscores the importance of tailoring molecular analyses to specific patient populations to optimize risk stratification and clinical decision-making [20]. Tumorigenesis in PTC, as in other cancers, results from genomic instability in somatic cells, which fosters the emergence of aggressive clones capable of thriving and outcompeting less fit cells within the tumor microenvironment. This evolutionary competition leads to variability in genomic composition, reflected in differences in AF. Despite this, the overall density of somatic mutations in PTC remains relatively low, which is thought to underlie its generally indolent clinical behavior [20].

### 4.6. Limitations

Our study has several limitations that warrant consideration. The small sample size (N = 34) restricts the statistical power and generalizability of our results. This limitation stems from the common clinical practice of avoiding biopsies for thyroid nodules < 1 cm, which likely reduced the pool of eligible cases. Because of our small sample size, to better contextualize the contribution of *BRAF V600E* AF, future research should compare our findings with analyses that assess BRAF V600E mutation status alone. This could help determine whether AF provides additional prognostic value beyond mutation presence. Additionally, the dual-center design, encompassing data from two urban Montreal hospitals, may introduce geographical selection bias, potentially limiting the applicability of findings to broader populations. Expanding future studies to include data from diverse geographical regions across Quebec or Canada could address this limitation. Furthermore, our analyses did not account for potential confounding factors known to influence MPTC progression, such as family history of thyroid cancer, tumor size, thyroid-stimulating hormone (TSH) levels, and obesity-related markers [20,30]. Controlling for these variables in future research will provide a more nuanced understanding of the interplay between *BRAF V600E* AF and tumor behavior. Despite these limitations, our findings contribute valuable insights into the potential role of *BRAF V600E* AF as a biomarker for MPTC aggressiveness. Larger, multicenter studies incorporating diverse populations and controlling for key confounders are needed to validate these results and refine clinical decision-making in the management of MPTC.

## 5. Conclusions

This study provides preliminary evidence that elevated *BRAF V600E* AF is associated with aggressive features in MPTC. A noted trend linking higher AF with lymph node metastasis highlights the need for further investigation to clarify its clinical significance. These findings suggest that *BRAF V600E* AF could serve as a marker of overall MPTC aggressivity and may be considered alongside other clinical and radiological findings to help determine a patient’s prognosis. If future studies on larger samples replicate our findings, *BRAF V600E* AF may prove to be a clinically useful biomarker to guide management decisions for MPTC, particularly in determining the appropriate balance between active surveillance and more aggressive interventions, such as hemi-thyroidectomy, total thyroidectomy, or limited central neck dissection. Such research is needed to help refine risk stratification and optimize patient-specific treatment strategies in MPTC management.

## Figures and Tables

**Table 1 cancers-17-02553-t001:** Sociodemographics and descriptives (N = 34).

Variable	Mean (SD) or N (%)
**Age**	49.6 (12.6)
**Biological sex**	
Female	31 (91.2%)
Male	3 (8.8%)
**Allele frequency**	19.2 (13.1)
**Aggressiveness**	
Non-aggressive	15 (44.1%)
Aggressive	19 (55.9%)
**Lymph node metastasis**	
No metastasis	24 (70.6%)
Metastasis	10 (29.4%)
**Aggressive histology**	
Not aggressive	27 (79.4%)
Aggressive	7 (20.6%)
**Extrathyroidal extension**	
No	31 (91.2%)
Yes	3 (8.8%)
**Lymphovascular invasion**	
No	33 (97.1%)
Yes	1 (2.9%)
**Bethesda score**	
B3	4 (11.8%)
B4	1 (2.9%)
B5	6 (17.6%)
B6	23 (67.7%)
**Histological subtype**	
Classical	19 (55.9%)
Follicular	6 (17.6%)
Oncocytic follicular	2 (5.9%)
Tall cell	7 (20.6%)

## Data Availability

Data can become available upon reasonable request to the corresponding author.

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
