# Peer review of "Association Between BRAF V600E Allele Frequency and Aggressive Behavior in Papillary Thyroid Microcarcinoma"

_cancers, 2025, doi:10.3390/cancers17152553_

Round 1
Reviewer 1 Report (Previous Reviewer 2)
Comments and Suggestions for Authors
The revised version of the manuscript is not improved. There are still issues that need clarification:
- the aim of the study is still not clear enough, the first and the second objective are the same; please clarify; what do you mean by “any histo-pathological determinant of MPTC tumor aggressiveness”? what do you mean by “individual histo-pathological determinants of MPTC tumor aggressiveness (lymph node metastasis, ETE, extensive LVI, extensive vascular invasions, or high-risk histological subtypes)”?
- the Material and Methods part is not well organized, the Pathological Data is mixed with the Statistical Analysis and appear everything in the Chapter Statistical Analysis which is not adequate; This part “Pathological diagnoses were based on the 2017 World Health Organization (WHO) classification of endocrine tumors. Cytology and final pathology samples were independently reviewed by board-certified, head and neck fellowship-trained pathologists.” does not belong to the Statistical Analysis;
- what do you mean by “extensive vascular invasion”? And what about its association with MPTCs?
- the Results part should be sub-divided in separate Chapters in order to highlight better the results. In the present form the information is not presented in a logical, comprehensive manner
- the Figures should be accompanied with a description of the results they highlight;
- the word “aggressive” and “aggressiveness” is still too much used in the article, the readers will remain with the idea that MPTCs are aggressive tumors, which is very false. Please address this issue throughout the entire manuscript;
- the main problem remains that the conclusion of the study is not supported by the results. The authors should perform major revision in the Results Chapter so that the information presented here appears clear and transparent, and it supports the conclusions;
- The article needs English revision
Author Response
Please see attachement.

Reviewer 2 Report (New Reviewer)
Comments and Suggestions for Authors
This is an interesting study investigating the role of BRAF V600E mutation and its allele frequency (AF) in the prognosis of Papillary thyroid microcarcinoma (MPTC). The manuscript has a good experimental design. The only problem is that the reader is confused by the positive or negative impact of BRAF V600E AF. A table may be more helpful for the positive and negative findings previously found. Since the findings are not very well supported by the small sample-size, the authors may compare these results with the results found by the analysis of BRAF V600E without the inclusion of AF, since it may be proved that if the outcome is the same the additional information of AF may not necessary.
Minor comments
Introduction
Lines 62-64 & 82-84; please include also manuscripts published by the journal including more data as hob-nail features ecc (Cancers 2022, 14(11), 2785; https://doi.org/10.3390/cancers14112785).
Line 238-239: did you mean ETE also here as you deal with this in the following paragraph?
Line 303: please delete ‘a’
Lines 303-304: ‘allele frequency’ has been already defined as AF
Author Response
Please see the attachment.

This manuscript is a resubmission of an earlier submission. The following is a list of the peer review reports and author responses from that submission.
Round 1
Reviewer 1 Report
Comments and Suggestions for Authors
The manuscript is well written, the hypothesis is logically structured. However, the biggest weakness of the research is the number of cases. Although the initial number of cases is quite large, in the end the number of samples meeting the selection criteria was only 34, which significantly reduces the value of the manuscript. This is precisely why the statistical calculations and the conclusions drawn from them are not strong enough. With a larger number of cases, the quality of the manuscript would increase significantly.
Reviewer 2 Report
Comments and Suggestions for Authors
In the present study, the authors aimed to evaluate the association between BRAF V600E AF and tumor aggressiveness in a series of MPTCs. The article addresses a very important clinical question (to provide critical insights, potentially informing clinical decisions regarding active surveillance versus surgical intervention and determining the extent of surgical management required – to identify that subset of MPTCs that needs additional treatment compared to cases for which active surveillance would be enough). However, the article needs major revisions. The following issues need clarifications:
- the aim of the study is not clear enough, the information is repeated (the first and the second objective are the same); also the next phrase is not clear (please revise the English);
- MPTCs are generally considered indolent tumors, with excellent prognosis. It is true that there is a subset of MPTCs that might behave more aggressively and be associated with adverse outcomes (like for ex. tumor recurrence). However, the word “aggressive” and “aggressiveness” is too much used in the article, the readers will remain with the idea that MPTCs are aggressive tumors, which is very false. Please address this issue throughout the entire manuscript;
- in routine practice we don’t generally have cases of MPTCs (tumors <1 cm) with extensive lymphovascular invasion (LVI) and extensive vascular invasion. Please explain why did you choose these variables or if not, please exclude them;
- what do you understand by “more extensive ETE”, pg 2, line 66?
- “insular cell” is not a histological type of PTC; if there are insular areas (poorly differentiated arears), these cases should be considered as poorly differentiated papillary thyroid carcinomas and not PTCs; please use the WHO (World Health Organization) Classifications of Tumors of the Thyroid Gland and the American Joint Committee on Cancer/Union for International Cancer Control (AJCC/UICC) TNM Classification of Tumors to define tumor histology and pathological stage;
- A new subchaper Statistical analysis should be added in the Material and Methods to better explain the statistical analysis (for example to explain how Alderson-Darling test was used);
- What about the follow-up. Do you have data regarding the outcomes? This would represent the best criteria to asses the prognostic value of BRAF V600E AF in your series of MPTCs
- The Results part should be more documented;
- The literature data is highly controversial regarding the prognostic value of BRAFV600E mutation in PTCs. However, the association of BRAFV600E mutation with other risk factors (eg. age≥55 years-old, male gender, conventional and tall cell histology, tumor size>40mm, extrathyroidal extension, multifocality and lymph node metastasis) appears to be a better predictor of adverse outcomes for PTC patients, compared to BRAFV600E mutation alone (doi: 10.3390/cancers15164053). With regard to this, what about your series of PTMCs? Please discuss this aspect.
- The article needs English revision
Comments on the Quality of English Language
The article needs English revision, in some aspects major revision (eg. the aim of the study)
Reviewer 3 Report
Comments and Suggestions for Authors
Interesting and methodologically well developed study. The small number certainly requires further studies, but these preliminary results may suggest a possible clinical application of this biomarker in predicting the aggressiveness of mPTC by influencing surgical decision making. The English is good. The bibliography is adequate. I would expand and verify only the captions of the 3 figures. In particular, it seems to me that the caption of figure 1 and figure 2 are inverted.
Round 2
Reviewer 2 Report
Comments and Suggestions for Authors
The revised version of the manuscript is improved. However, there are still issues that need clarification:
- the aim of the study is still not clear enough, the first and the second objective are the same; please clarify;
- according to the newest version of Thyroid Tumors Classification and Staging TNM 2016, extra thyroidal extension is considered macroscopically evidence of tumor extension into the striated muscle of the neck, there are no terms as “extensive” or ‘’more extensive’’ extra thyroidal extension; please do not use them and clarify this issue throughout the article; microscopic extrathyroidal extension (mETE) refers to extension into peri-thyroidal adipose tissue? Please clarify;
- the word “aggressive” and “aggressiveness” is still too much used in the article, please address this issue throughout the entire manuscript;
